# Implementing a Substance-Use Screening and Intervention Program for People Living with Rifampicin-Resistant Tuberculosis: Pragmatic Experience from Khayelitsha, South Africa

**DOI:** 10.3390/tropicalmed7020021

**Published:** 2022-01-31

**Authors:** Anja Reuter, Buci Beko, Boniwe Memani, Jennifer Furin, Johnny Daniels, Erickmar Rodriguez, Hermann Reuter, Lize Weich, Petros Isaakidis, Erin von der Heyden, Yulene Kock, Erika Mohr-Holland

**Affiliations:** 1Médecins Sans Frontières, Khayelitsha 7784, South Africa; msfocb-khayelitsha-tbconsl@brussels.msf.org (B.B.); MSFOCB-khayelitsha-psu@brussels.msf.org (B.M.); MSFOCB-Khayelitsha-Tbdata@brussels.msf.org (J.D.); psi.mar.rodriguez@gmail.com (E.R.); or msfocb-khayelitsha-drtb-epi@brussels.msf.org (E.M.-H.); 2Department of Global Health and Social Medicine, Harvard Medical School, Boston, MA 02115, USA; jenniferfurin@gmail.com; 3Faculty of Health Sciences, University of Cape Town, Cape Town 7701, South Africa; hermannreuter@gmail.com; 4Department of Psychiatry, Faculty of Medicine and Health Sciences, Stellenbosch University, Cape Town 7701, South Africa; lizew@sun.ac.za; 5Médecins Sans Frontières Southern Africa Medical Unit, Cape Town 7925, South Africa; Petros.Isaakidis@joburg.msf.org; 6Provincial Government of the Western Cape, Cape Town 8000, South Africa; Erin.vonderHeyden@westerncape.gov.za; 7National Department of Health Tuberculosis Program, Pretoria 0187, South Africa; Yulene.Kock@health.gov.za

**Keywords:** RR-TB, substance use, integrated care, person-centered care, loss-to-follow-up, ASSIST, brief intervention, SBIRT

## Abstract

Substance use (SU) is associated with poor rifampicin-resistant tuberculosis (RR-TB) treatment outcomes. In 2017, a SBIRT (SU screening-brief intervention-referral to treatment) was integrated into routine RR-TB care in Khayelitsha, South Africa. This was a retrospective study of persons with RR-TB who were screened for SU between 1 July 2018 and 30 September 2020 using the ASSIST (Alcohol, Smoking and Substance Involvement Screening Test). Here we describe outcomes from this program. Persons scoring moderate/high risk received a brief intervention and referral to treatment. Overall, 333 persons were initiated on RR-TB treatment; 38% (*n* = 128) were screened for SU. Of those, 88% (*n* = 113/128) reported SU; 65% (*n* = 83/128) had moderate/high risk SU. Eighty percent (*n* = 103/128) reported alcohol use, of whom 52% (*n* = 54/103) reported moderate/high risk alcohol use. Seventy-seven persons were screened for SU within ≤2 months of RR-TB treatment initiation, of whom 69%, 12%, and 12% had outcomes of treatment success, loss to follow-up and death, respectively. Outcomes did not differ between persons with no/low risk and moderate/high risk SU or based on the receipt of naltrexone (*p* > 0.05). SU was common among persons with RR-TB; there is a need for interventions to address this co-morbidity as part of “person-centered care”. Integrated, holistic care is needed at the community level to address unique challenges of persons with RR-TB and SU.

## 1. Introduction

Rifampicin-resistant tuberculosis (RR-TB) affects half a million people each year and is associated with a high rate of morbidity and mortality [1]. Although there have been recent therapeutic improvements leading to better treatment outcomes, global success rates are just over 61% [2]. Treatment for RR-TB is characterized by a long duration and high pill burden, and multiple patients are unable to adhere to this grueling therapy or complete the prescribed course of treatment [3,4,5,6]. People who stop taking treatment early or who miss more than 8 weeks of therapy are given an outcome of Loss to Follow-Up (LTFU). Rates of LTFU vary considerably between settings and are driven by multiple factors, including the use of alcohol and other substances [7,8,9]. 

Substance-use disorder (SUD) is a common co-morbid condition among people living with RR-TB. It is a risk for poor treatment outcomes, including LTFU [10]. Substance use (SU) also complicates other aspects of RR-TB treatment including overlapping toxicities with TB treatment [11], increased experiences of stigma, and lack of access to SU treatment given the infectious nature of RR-TB. 

The World Health Organization (WHO) has recommended that health service providers offer screening and interventions for SU within a framework called Screening, Brief Intervention, and Referral to Treatment or SBIRT [12]. The WHO also recommends naltrexone [13], an opioid-receptor blocker that has been shown to reduce alcohol use and decrease the risk of relapse [14], be offered as a pharmacotherapeutic option for persons with an alcohol use disorder. Given the high rates of SU seen among people with RR-TB and the potential health consequences of such use, both a SBIRT model and access to naltrexone for persons with alcohol use disorder could be helpful to integrate into outpatient RR-TB care. In 2017, the medical humanitarian organization Médecins Sans Frontières (MSF) implemented a pilot SBIRT program in collaboration with the Department of Health for people living with RR-TB in Khayelitsha, South Africa. Here we aim to describe the results of this pilot program by describing SU screening, screening scores, and RR-TB treatment outcomes for persons screened for SU within 2 months of treatment initiation. 

## 2. Materials and Methods

### 2.1. Study Design and Participants

This was a retrospective cohort study of people treated for RR-TB in Khayelitsha, South Africa between 1 July 2018 and 30 September 2020. Persons were considered eligible for the cohort study if they were initiated on RR-TB treatment during that time period and if they underwent an initial screening for SU at any time during the treatment. Only those individuals who underwent screening within the initial 2 months of RR-TB treatment were included in the treatment outcome analysis in order to assess the potential impact of a timely SBIRT intervention on treatment response. 

### 2.2. Setting

This study was conducted in Khayelitsha, a peri-urban township located on the outskirts of Cape Town (in the Western Cape province) which is home to approximately 400,000 residents, a majority of whom live in poverty [15]. TB, RR-TB, and HIV are significant health problems in the population; the RR-TB case notification rate is 55/100,000/year, and approximately 70% of RR-TB patients are HIV co-infected [16]. A survey conducted by the Western Cape Government describes that 62% of Khayelitsha residents drink alcohol, with 42% having had contact with police in the previous 6 months due to alcohol use [17]. Reported rates of illicit substances such as TIK (crystal methamphetamines), MANDRAX (250 mg methaqualone and 25 mg diphenhydramine), and “woonga” (a heroin-based cocktail drug) are the highest in South Africa with 7.1% of respondents from the Western Cape reporting previous illicit drug use in the past 3 months in a population survey [18]. MSF has been supporting the Department of Health in the decentralized management of RR-TB in 10 health clinics in Khayelitsha, South Africa for more than 10 years. LTFU remains a significant challenge ranging from 20–25% in Khayelitsha since 2017, despite implementation of shorter treatment regimens. 

### 2.3. Screening, Brief Intervention, and Referral to Treatment (SBIRT) Description

The SBIRT package, summarized in Figure 1, was integrated into routine primary health care clinic visits; it was usually conducted within a single clinic visit, although sometimes two if requested by the patient, and it was most commonly administered by a trained lay RR-TB counselor. For some individual patients it was administered by the doctor or nurse working in the TB room. Training and mentoring on the screening and brief intervention tool was provided by MSF staff to RR-TB counselors, as well as to TB doctors and nurses. Most department of health staff also received motivational interviewing training facilitated by MSF through an external partner. In addition to training and mentoring, MSF provided additional human resources to support the program at the 10 clinics with an MSF RR-TB counselor, and for a shorter period of time with an MSF nurse. 

#### 2.3.1. Screening

The screening component of the SBIRT was done using Alcohol, Smoking and Substance Involvement Screening Test (ASSIST) [12]. An overall SU score is given as a score for each individual substance assessed. Notably, as part of this intervention, if a person reported on the initial question that they had “never” used any substances in their lifetimes, then no additional screening questions were asked. Persons who reported no lifetime SU, or scored low risk for SU received no further intervention. Persons who scored moderate or high risk received a brief intervention and referral to treatment. 

#### 2.3.2. Brief Intervention

The brief intervention was based on the WHO ASSIST linked brief intervention [12], however the intervention was tailored to persons with HIV/TB. 

#### 2.3.3. Referral/Treatment

The referral/treatment component included multiple aspects. First, people were provided with information and facilitated access (including transport) to a RR-TB specific SU support group and the City of Cape Town Matrix^®^ outpatient rehabilitation program. Second, details of the Khayelitsha Narcotics Anonymous group and the contact details of a RR-TB counselor experienced in SU counseling were shared (both verbally and in health information and educational materials). Third, persons thought to require inpatient rehabilitation support were referred to the doctor to discuss RR-TB care at the central hospital, where an inpatient SU program was available. Finally, persons using alcohol were offered the opportunity to see a medical practitioner for naltrexone pharmacotherapy initiation. 

Naltrexone was offered to all patients who had moderate or high-risk alcohol use, and who did not have a medical contra-indication to its use. Therapy was initiated if the patient expressed a goal to reduce the use of alcohol and if they indicated that they wanted to start therapy. Initially, patients were issued a 1-month supply. Although counseling was provided that the ideal way of taking medication was daily, the “Sinclair method” was proposed as an alternative [19]. This method entails the use of naltrexone at times when the risk of drinking is high—for example, at weekends. 

### 2.4. Follow Up Care 

During the follow-up clinical RR-TB visits post-SBIRT implementation, discussions around SU using the motivational interviewing techniques could continue between the TB care provider and the patient in order to increase motivation to reduce harmful SU. 

### 2.5. Outcome Measures and Definitions

Patients were defined as never using substances if they reported that they never used alcohol, tobacco, and drugs in their lifetime. Persons who reported ever using substances were given a risk score (low, moderate, or high risk) for each substance of use according to the criteria outlined in the ASSIST [12]. Persons were given an overall risk classification defined as the highest risk classification reported for a single substance of use for each person (some persons reported high risk use of multiple substances). RR-TB treatment outcomes were defined according to the standard WHO definitions [20]; cure and treatment completed were combined as treatment success. 

### 2.6. Data Collection and Analysis 

Data were obtained from the South African Electronic Drug-Resistant Tuberculosis Register (EDRWeb), paper RR-TB registers, and patients’ medical records, where the ASSIST and brief intervention tools were recorded. Data entered in EDRWeb were validated using information in patient medical records and RR-TB registers. Data on SU screening included the following variables: date of SU screening, SU screening results (substance/s of use and risk classification scores), date of the brief intervention, whether naltrexone was initiated, and if the patient experienced any serious adverse event while on naltrexone therapy. These data were entered into a RedCap Database [21] and linked to the routine programmatic data collected from exports from the routine monitoring systems. 

Population comparisons for baseline clinical and demographic variables were compared using chi square and Fisher and Wilcoxon rank-sum tests. For stratified analyses based on overall SU risk, patients were grouped as having no- or low-risk SU and moderate- or high-risk SU as this determined who had access to a brief intervention and referral to treatment. Categorical variables are presented as numbers and proportions and continuous variables are presented as medians and interquartile ranges. Descriptive statistics were used to report on the SU screening cascade, including risk scores for substances used and referral to pharmacotherapy, and RR-TB treatment outcomes. RR-TB outcomes for persons screened for SU within less than or equal to 2 months of RR-TB treatment initiation were reported stratified by no/low risk SU versus moderate/high risk SU. Finally, for persons with moderate- and high-risk alcohol use who were screened for SU within less than or equal to 2 months of RR-TB treatment initiation, RR-TB outcomes were reported stratified by receipt of naltrexone. *p*-values < 0.05 were considered statistically significant. 

### 2.7. Ethics

Ethical approval was obtained from the University of Cape Town Human Research Ethics Committee (HREC 499/2011) for this study. Additionally, this research fulfilled the exemption criteria set by the MSF Ethical Review Board for a posterior analysis of routinely collected clinical data and thus did not require MSF ERB review. 

## 3. Results

### 3.1. Clinical and Demographic Characteristics Based on Substance-Use (SU) Screening

Over the study period, 333 persons were initiated on RR-TB treatment, of whom 128 (38%) were screened for SU using the ASSIST tool. Table 1 compared the clinical and demographic characteristics between those screened and those not screened. Of note, a significantly higher proportion of people initiated on treatment in the hospital did not undergo SU screening. 

### 3.2. Substance Use Screening: Substances of Use and Risk Classification

Of the 128 persons screened for SU, 88% (*n* = 113/128) reported using substances. The median time from RR-TB treatment initiation to SU screening was 1.3 months (interquartile range [IQR] 0.5–3.5). Of persons screened, 65% (83/128) reported moderate- or high-risk SU. Overall, 87% (*n* = 72/83) of those with moderate- or high-risk SU received a brief intervention. 

The reported substances of use (not mutually exclusive) and risk classification based on the ASSIST screening can be seen in Figure 2; the median number of substances used was 2 (IQR 1–2), with 61% (*n* = 78/128) reporting >1 substance of use. Alcohol was the most frequently reported substance of use, with 80% (*n* = 103/128) of those screened reporting alcohol use and 52% (*n* = 54/103) reporting moderate- or high-risk alcohol use. Tobacco followed alcohol as the second most common substance of use with 65% (*n* = 83/128) of persons screened reporting tobacco use and 39% (*n* = 65/83) reporting moderate- or high-risk tobacco use. 

Among the 54 persons with moderate/high risk alcohol use, 17 (31%) were initiated on naltrexone pharmacotherapy. None of the 17 persons initiated on naltrexone experienced any serious adverse events while still on therapy. 

### 3.3. Clinical and Demographic Characteristics Based on Highest Risk Classification

Differences in the clinical and demographic characteristics based on the highest overall risk classification determined from the SU screening are shown in Table 2. Men, those 30–39 years of age, and those on antiretroviral therapy were significantly more like to have moderate/high risk SU (*p* < 0.05). 

### 3.4. Rifampicin-Resistant Tuberculosis (RR-TB) Treatment Outcomes among Patients Screened for Substance Use Disorder within 2 Months of RR-TB Treatment Initiation

Of the 128 persons screened for SU, 77 were screened within 2 months of RR-TB treatment initiation and included in the outcome analysis. In total, 42% (*n* = 32/77) had no- or low-risk SU and 58% (*n* = 45/77) had moderate- or high-risk SU. Of the 77 persons, 69%, 12%, and 12% had outcomes of treatment success, LTFU, and death, respectively. There were no significant differences in treatment outcomes between persons with no- or low-risk SU and those with moderate- or high-risk SU (*p* > 0.05, Table 3).

Of the 77 persons who received SBIRT and were included in the analysis of outcomes, 33 were eligible for naltrexone (had moderate or high risk SU). Of those, 11 were initiated on naltrexone therapy (Table 4). There were no significant differences in treatment outcomes based on whether or not naltrexone was received among this group (*p* > 0.05, Table 4). 

## 4. Discussion

We report the results of the implementation of an SBIRT intervention among a cohort of people living with RR-TB in Khayelitsha receiving care in the primary health care setting between 2018 and 2020. Of the 333 people who started RR-TB treatment in Khayelitsha during the study period, just over a third (128) received this SBIRT intervention. While it was the intention that the services be offered to everyone, the programmatic conditions meant that most people with RR-TB did not receive the intervention. Persons initiated on treatment in Khayelitsha at the primary health-care level were more likely to have received the SBIRT compared to persons initiating treatment at the hospital level, due to the fact that this SBIRT package was only implemented at primary health-care facilities.

The low coverage of SBIRT reflects some of the implementation challenges associated with this program. The majority of the screening was done by RR-TB counselors who moved between 10 primary health care facilities. These counselors sometimes experienced logistical challenges when coordinating being at the clinic at the same time as the patient’s routine clinical appointment. This highlights the importance of having staff stationed at primary health care facilities trained on SBIRT so as to minimize missed opportunities at routine RR-TB clinical visits [22]. Although the TB nurses stationed at the primary health-care facilities were also trained on the SBIRT intervention, few administered this intervention due to competing clinical priorities, and lack of a private space to consult with the patient. This highlights the importance of the SBIRT package being implemented with adequate human resources and appropriate cadres of staff [23,24,25]. In our setting we found that lay counselors had fewer competing clinical priorities than nurses or doctors and were key in the implementation of these SBIRT services. In other settings, community health workers have also been used successfully to implement SBIRT and are a promising cadre of staff to involve in substance-use services [26]. 

Overall, reported rates of SU were high among those screened, with more than 60% of participants who underwent screening reporting moderate- or high-risk SU. This rate is higher than that reported in other cohorts [27]. The most common substances used were alcohol and tobacco. Moderate-/high-risk SU was more common among males, persons between the ages of 30–39 years, and persons on antiretroviral treatment; however the latter association is likely to be a statistical artifact. Although the high rates of SU seen in this cohort suggest SBIRT should be offered to all persons undergoing treatment for RR-TB, urgent implementation could be prioritized among these groups. 

Our study did not reveal any statistically significant differences in RR-TB treatment outcomes when stratified by SU risk score. There may be several reasons for this, including the relatively small sample size and the relatively ubiquitous nature of SU reported among those screened. It may also be that the SBIRT intervention component had some impact on the ability of persons to remain engaged in care, although this would need to be assessed in a more formal study. Supporting this hypothesis, we note that the rates of treatment success in our cohort are 69%, higher than those historically reported nationally and in the Khayelitsha setting [1,16,28]. Some of this may be due to the expanded use of the shorter and more effective treatment regimens that have been rolled out in South Africa. However, the fact that such improved outcomes were also registered among persons with moderate/high SU in our cohort is encouraging. We do note that among those who reported moderate/high-risk SU, rates of LTFU remained unacceptably high at 16%. While not reaching statistical significance, the difference in LTFU rates among those with moderate/high risk SU compared to no/low substance use supports the fact that more work is needed to identify and support persons with comorbid RR-TB and substance use disorders. 

We aimed to assess the use of naltrexone therapy as possible therapeutic intervention in those persons with moderate-/high-risk alcohol use on RR-TB treatment outcomes, as this has not previously been described [29]. However, uptake in our study was low with approximately one third of those who were eligible accessing therapy. One reason for this may be that naltrexone was available only through a medical practitioner and thus required an extra visit where the person conducting the SBIRT was a counselor. When implementing such interventions in future it will be important to ensure that naltrexone is available at the same time that the SBIRT is conducted. Also, persons with RR-TB are on a number of medications; additional tablets may be less acceptable to persons than those in other contexts on fewer concomitant medications. The low uptake greatly limited our ability to assess the utility of naltrexone use and could account for the fact that we did not see any significant association between naltrexone use and treatment outcomes. It was encouraging, however, that there were no serious adverse events reported among people on RR-TB treatment while on naltrexone, although more rigorous data is needed. 

This retrospective study had a number of important limitations. Firstly, there was a small sample size, with only 38% of persons treated for RR-TB accessing this intervention. This impacted the ability to detect significant differences between the various groups of RR-TB patients evaluated, and may have led to selection bias. Despite efforts to systematically offer SBIRT to all persons with RR-TB, health center staff may have been more likely to offer this intervention to persons with SU challenges, which may have resulted in a higher rate of SU seen in our cohort than in the overall population of people started on RR-TB treatment in Khayelitsha during the study period. Also, persons with a more significant alcohol use disorder are likely to have preferentially received naltrexone therapy and this may have influenced the treatment outcomes observed. Additionally, persons in hospitals, or persons with RR-TB with early mortality were less likely to receive the SBIRT package, which may have led to bias in treatment outcome results. Finally, this analysis was conducted in a single district which may limit the generalizability of our findings.

## 5. Conclusions

In spite of these limitations, our study has several important findings. First, SU is extremely common among persons with RR-TB and there is a need for interventions to screen for and address this co-morbidity as part of “person-centered care”. Second, while a SBIRT intervention can be implemented in the outpatient setting among people living with RR-TB, there is a need for dedicated staff and resources to do this work in order to reach the entire population. Third, among people with SU, good treatment outcomes can be achieved—although rates of LTFU still remain unacceptably high among those with moderate or high-risk SU scores. Integrated, holistic care is needed at the community level for persons with RR-TB and our study shows that such care needs to include services to addresses the unique challenges that persons with RR-TB who use substances face. 

## Figures and Tables

**Figure 1 tropicalmed-07-00021-f001:**
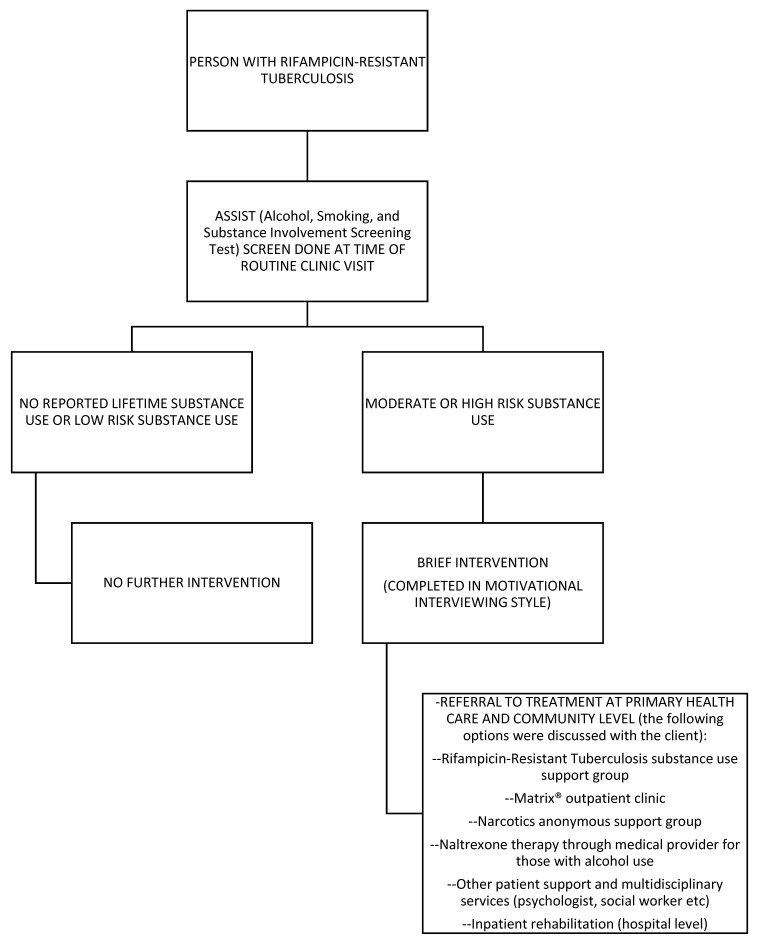
Rifampicin-resistant tuberculosis substance-use intervention flow diagram (Screening, Brief Intervention and Referral to Treatment Model). This intervention could be implemented by doctors, nurses, or trainer lay counselors at primary health care facility.

**Figure 2 tropicalmed-07-00021-f002:**
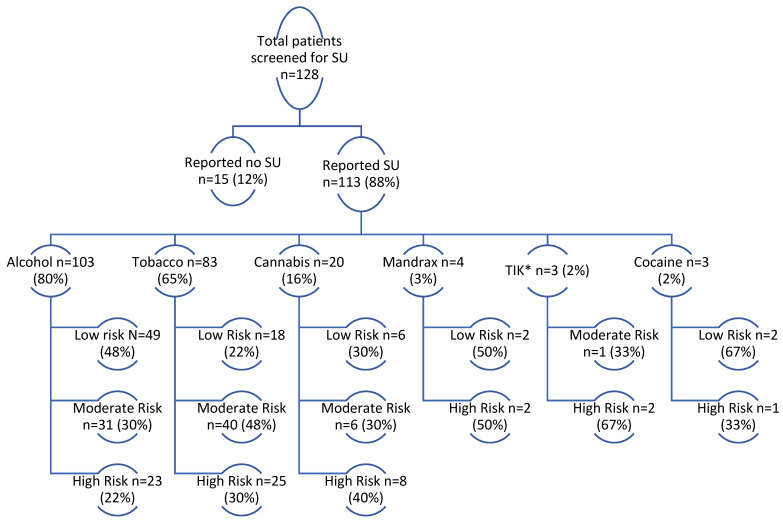
Substance-use screening outcomes among persons initiated on RR-TB treatment from July 2018-September 2020. * TIK= crystal methamphetamines.

**Table 1 tropicalmed-07-00021-t001:** Clinical and demographic characteristics for persons initiated on rifampicin-resistant tuberculosis (RR-TB) treatment from July 2018–September 2020 stratified by whether substance-use screening was conducted.

	Total *n* = 333	Not Screened *n* = 205	Screened *n* = 128	*p*-Value
Male	195 (58.6)	117 (57.1)	78 (60.9)	0.49
Median Age, years	34 (28–42)	34 (28–42)	35 (29–43)	0.58
Age Category, years				
<20	19 (5.7)	10 (4.9)	9 (7.0)	
20–29	87 (26.1)	58 (28.3)	29 (22.7)	
30–39	119 (35.7)	72 (35.1)	47 (36.7)	
40–49	66 (19.8)	38 (18.5)	28 (21.9)	
≥50	42 (12.6)	27 (13.2)	15 (11.7)	0.69
Disease classification				
Xpert MTB/RIF unconfirmed	34 (10.2)	25 (12.2)	9 (7.0)	
Rifampicin-mono resistance	77 (23.1)	53 (25.9)	24 (18.8)	
MDR including injectable resistance	191 (57.4)	113 (55.1)	78 (60.9)	
MDR plus fluroquinolone resistance	31 (9.3)	14 (6.8)	17 (13.3)	0.051
Previous TB treatment history None	143 (42.9)	90 (43.9)	53 (41.4)	
Previous 1st line TB treatment	162 (48.7)	96 (46.8)	66 (51.6)	
Previous 2nd line TB treatment	28 (8.4)	19 (9.3)	9 (7.0)	0.64
Disease Site				
Pulmonary TB	314 (94.3)	194 (94.6)	120 (93.8)	
Extra-Pulmonary TB	19 (5.7)	11 (5.4)	8 (6.3)	0.74
Site of Treatment initiation	288 (86.5)	167 (81.5)	121 (94.5)	
Primary Health Care Facility Hospital	45 (13.5)	38 (18.5)	7 (5.5)	0.001 *
HIV Positive	226 (67.9)	138 (67.3)	88 (68.8)	0.79
Median CD4 count	79 (28–239) ^	73 (24–217) ^	102 (35–247) ^	0.30
On Antiretroviral Therapy	218 (96.5)	132 (95.7)	86 (97.7)	0.38

* Indicates statistical significance. ^ 30, 23, and 7 persons missing CD4 counts in the total, not screened and screened groups, respectively. Data are presented as number and proportions or medians and interquartile ranges.

**Table 2 tropicalmed-07-00021-t002:** Clinical and demographic characteristics of persons initiated on RR-TB treatment from July 2018–September 2020 who were screened for substance use, stratified by no- or low-risk SU and moderate- or high-risk SU.

	Total Screened *n* = 128	No-/Low-Risk SU *n* = 45	Moderate-/High-Risk SU *n* = 83	*p*-Value
Male	78 (60.9)	14 (31.1)	64 (77.1)	<0.001 *
Median Age, years	35 (29–43)	34 (25–47)	35 (30–42)	0.64
Age Category, years				
<20 20–29 30–39 40–49	9 (7.0) 29 (22.7) 47 (36.7) 28 (21.9)	5 (11.1) 13 (28.9) 11 (24.4) 6 (13.3)	4 (4.8) 16 (19.3) 36 (43.4) 22 (26.5)	
>=50	15 (11.7)	10 (22.2)	5 (6.0)	0.006 *
Time to SU screening				
<=2 months	77 (60.2)	32 (71.1)	45 (54.2)	
>2 months	51 (39.8)	13 (28.9)	38 (45.8)	0.062
Disease classification				
Xpert MTB/RIF unconfirmed Rifampicin-mono resistance MDR including injectable resistance	9 (7.0) 24 (18.8) 78 (60.9)	3 (6.7) 8 (17.8) 25 (55.5)	6 (7.2) 16 (19.3) 53 (63.9)	
MDR plus fluroquinolone resistance	17 (13.3)	9 (20.0)	8 (9.6)	0.43
Previous TB treatment history				
None Previous 1st line TB treatment	53 (41.4) 66 (51.6)	21 (46.7) 20 (44.4)	32 (38.6) 46 (55.4)	
Previous 2nd line TB treatment	9 (7.0)	4 (8.9)	5 (6.0)	0.48
Disease Site				
Pulmonary TB	120 (93.8)	42 (93.3)	78 (94.0)	
Extra-pulmonary TB	8 (6.3)	3 (6.7)	5 (6.0)	0.89
Site of Treatment initiation				
Primary Health Care Facility	121 (94.5)	41 (91.1)	80 (96.4)	
Hospital	7 (5.5)	4 (8.9)	3 (3.6)	0.21
HIV Positive	88 (68.8)	29 (64.4)	59 (71.1)	0.44
Median CD4 count	102 (35–247) ^	100 (35–238)	107 (38–257) ^	0.95
On Antiretroviral Therapy	86 (97.7)	27 (93.1)	59 (100.0)	0.041 *

* Indicates statistical significance. ^ 7 and 7 persons missing CD4 counts in the total screened and moderate-/high-risk SU groups, respectively. Data are presented as number and proportions or medians and interquartile ranges.

**Table 3 tropicalmed-07-00021-t003:** RR-TB treatment outcomes for persons started on treatment from July 2018–September 2020 who were screened for substance use within 2 months of treatment initiation, stratified by no- or low-risk SU and moderate- or high-risk SU.

	Overall *n* = 77 *n* (%)	No-/Low-Risk SU *n* = 32 *n* (%)	Moderate/High Risk *n* = 45 *n* (%)
Treatment Success	53 (68.8)	21 (65.6)	32 (71.1)
Loss to Follow-up	9 (11.7)	2 (6.2)	7 (15.6)
Died	9 (11.7)	5 (15.6)	4 (8.9)
Failed by Treatment	3 (3.9)	3 (9.4)	0 (0)
Not Evaluated	3 (3.9)	1 (3.1)	2 (4.4)

Data are presented as numbers and proportions.

**Table 4 tropicalmed-07-00021-t004:** RR-TB treatment outcomes for persons started on treatment from July 2018–September 2020 with moderate- or high-risk alcohol use who were screened for substance use within 2 months of treatment initiation, stratified by receipt of naltrexone.

	Overall *n* = 33 *n* (%, Column)	Did not Receive Naltrexone *n* = 22	Received Naltrexone *n* = 11
Treatment Success	23 (69.7)	16 (72.7)	7 (63.6)
Loss to Follow-up	6 (18.2)	4 (18.2)	2 (18.2)
Died	2 (6.1)	1 (4.5)	1 (9.1)
Not Evaluated	2 (6.1)	1 (4.5)	1 (9.1)

Data are presented as numbers and proportions.

## Data Availability

Data is available from the corresponding author upon request.

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
