# Peer review of "Implementing a Substance-Use Screening and Intervention Program for People Living with Rifampicin-Resistant Tuberculosis: Pragmatic Experience from Khayelitsha, South Africa"

_tropicalmed, 2022, doi:10.3390/tropicalmed7020021_

Round 1
Reviewer 1 Report
It is an interesting and important study. On cannot find out statistically significant differences due to small sample size. However, it should not minimize the value of the research done. Looking forward to see the article with higher number of patients enrolled. The National TB Programme has to work on the improving of SBIRT coverage in order to increase the number of patients eligible to be enrolled.
Some minor comments to be considered by authors:
- Table #1. The row “Previous TB treatment history”- the total number of patients came up to 332 patients, not 333. If one patient does not match any treatment history, then it should be indicated with apostrophe and explanation should be provided on the bottom of the table.
- Line 219- “… and 58% (n=32/77)” should be checked and recalculated, if needed.
- Line 256, I guess it should be “than” not “then”
Author Response
Reviewer 1
Reviewer Comment
It is an interesting and important study. On cannot find out statistically significant differences due to small sample size. However, it should not minimize the value of the research done. Looking forward to see the article with higher number of patients enrolled. The National TB Programme has to work on the improving of SBIRT coverage in order to increase the number of patients eligible to be enrolled.
Authors Response
Thank you for your review and for your recognition of the importance of this work despite the limitations regarding the small sample size. We agree with you that the National TB Programme should roll this program out more broadly, and it is our hope that by publishing this work it can be used in advocacy to leverage policy change. We have mentioned this in the discussion section.
Reviewer Comment
Table #1. The row “Previous TB treatment history”- the total number of patients came up to 332 patients, not 333. If one patient does not match any treatment history, then it should be indicated with apostrophe and explanation should be provided on the bottom of the table.
Authors Response
Thank you for bringing this to our attention. The individual was not previously treated for TB; this has been amended in Table 1.
Reviewer Comment
Line 219- “… and 58% (n=32/77)” should be checked and recalculated, if needed.
Authors Response
We are sorry about this error. This was supposed to read as “58% (n=45/77)”. This has been corrected in the revised manuscript.
Reviewer Comment
Line 256, I guess it should be “than” not “then”
Authors Response
We are sorry about this error which has been corrected in the revised version of this manuscript.
Reviewer 2 Report
General
This paper explores the programmatic experience of implementing substance use screening and intervention in patients living with rifampicin-resistant tuberculosis in Khayelitsha, SA and how this affects treatment outcomes. During a period of over 2 years only 38% (128/333) of RR-TB patients initiated on treatment were screened for substance use. Outcomes could only be assessed for 77/128 (23%) patients who had been screened within 2 months of treatment initiation. Substance use, especially alcohol, was very common. No difference in treatment outcome was noted between no/low and medium/high risk users, nor with intervention of naltrexone for alcohol use; all likely due to the limited sample size. The authors (and MSF in general) are to be commended for their efforts to document how common substance use is in their SA setting, as well as highlighting the challenges of implementing an intervention to tackle this problem. They elaborately discuss these challenges and suggest remedies to improve screening and intervening. They also enumerate the limitations of their study design clearly in the discussion and give an excellent and concise conclusion. The paper is well written. Nevertheless, in my opinion I would rather see these data in a letter or correspondence format rather than an original research article. Specific remarks - abstract: the study of objective is missing in the abstract as well a sin the introduction. It is specified for the first time in the methods (section 2.1) - results: table 2: it is counter-intuitive to report pat on ART at increased risk for SU whereas HIV serology status is not correlated. The current analysis provides a statically significant result but I’m not convinced it makes sense from a clinical perspective.Author Response
Reviewer 2
Reviewer Comment
This paper explores the programmatic experience of implementing substance use screening and intervention in patients living with rifampicin-resistant tuberculosis in Khayelitsha, SA and how this affects treatment outcomes. During a period of over 2 years only 38% (128/333) of RR-TB patients initiated on treatment were screened for substance use. Outcomes could only be assessed for 77/128 (23%) patients who had been screened within 2 months of treatment initiation. Substance use, especially alcohol, was very common. No difference in treatment outcome was noted between no/low and medium/high risk users, nor with intervention of naltrexone for alcohol use; all likely due to the limited sample size. The authors (and MSF in general) are to be commended for their efforts to document how common substance use is in their SA setting, as well as highlighting the challenges of implementing an intervention to tackle this problem. They elaborately discuss these challenges and suggest remedies to improve screening and intervening. They also enumerate the limitations of their study design clearly in the discussion and give an excellent and concise conclusion. The paper is well written. Nevertheless, in my opinion I would rather see these data in a letter or correspondence format rather than an original research article
Authors Response
Thank you for your thoughtful reflections. We agree that this study is limited by the small sample size as this paper reflects the programmatic reality of the cohort that we work with. Additionally, given the paucity of data on substance use in RR-TB cohorts, this paper provides an important description of the model of care and implementation results. However, we would be unable, to convey the nuances of the population involved, the details of intervention provided, the complex methods used in the analysis, nor the context of the study results within a letter. The word limitations of a letter would also not allow for communication of strengths and limitations of the work in a meaningful way that readers could follow. We therefore are asking for consideration of the manuscript as an original research study.
Reviewer Comment
Abstract: the study of objective is missing in the abstract as well as in the introduction. It is specified for the first time in the methods (section 2.1)
Authors Response
Thank you for bringing this to our attention. As per your recommendation, we have added details regarding the aim in both the abstract and introduction section of the paper. The abstract now states: “Here we describe outcomes from this program.” The introduction section now states: “Here we aim to describe the results of this pilot program by describing SU screening, screening scores, and RR-TB treatment outcomes for persons screened for SU within 2-months of treatment initiation.”
Reviewer Comment
Results: table 2: it is counter-intuitive to report pat on ART at increased risk for SU whereas HIV serology status is not correlated. The current analysis provides a statically significant result but I’m not convinced it makes sense from a clinical perspective.
Authors Response
We are sorry about the confusion that this may have cause; we believe that this is merely a statistical artifact. We have mentioned this in the discussion section of the paper for clarity, which now states: “Moderate/high risk SU was more common among males, persons between the ages of 30-39 years, and persons on antiretroviral treatment; however, the latter is assumed to be a statistical artifact.”
Reviewer 3 Report
The authors used a descriptive study to identify substance use and other characteristics among persons with rifampicin-resistant tuberculosis in South Africa. Because substance use is associated with poor outcome, it may be important identify the characteristic for improvement of treatment. This article showed characteristics after screening of substance use, but there are some of errors which should be revised.
First, as there is not description of meaning of the numbers (e.g. mean, percentage, range, IQR and SD), it is difficult to understand the table 1. Authors should add the explanation of the numbers in table 1. Moreover, the description “^30 missing 235missing ^7 missing CD4” may be footnote of table 1. Authors should revise the table 1.
Second, as there are some of typo in the manuscript (e.g. [[1] in L40, “counsellors, as” in L97), authors should revise the manuscript.
Author Response
Reviewer 3
Reviewer Comment
The authors used a descriptive study to identify substance use and other characteristics among persons with rifampicin-resistant tuberculosis in South Africa. Because substance use is associated with poor outcome, it may be important identify the characteristic for improvement of treatment. This article showed characteristics after screening of substance use, but there are some of errors which should be revised.
Authors Response
Thank you for your careful review of our paper.
Reviewer Comment
First, as there is not description of meaning of the numbers (e.g. mean, percentage, range, IQR and SD), it is difficult to understand the table 1. Authors should add the explanation of the numbers in table 1. Moreover, the description “^30 missing 235missing ^7 missing CD4” may be footnote of table 1. Authors should revise the table 1.
Authors Response
Thank you for bringing this to our attention. We have added a legend to Table 1 clarifying these details.
Reviewer Comment
Second, as there are some of typo in the manuscript (e.g. [[1] in L40, “counsellors, as” in L97), authors should revise the manuscript.
Authors Response
We apologize for these errors. We have corrected these and have reviewed the entire manuscript for additional errors.
Round 2
Reviewer 2 Report
Thank you for the revision and additional comments.